# Incidence of Hospitalisation and Emergency Department Visits for Pneumococcal Disease in Children, Adolescents, and Adults in Liguria, Italy: A Retrospective Analysis from 2012–2018

**DOI:** 10.3390/vaccines10091375

**Published:** 2022-08-24

**Authors:** Matteo Astengo, Chiara Paganino, Daniela Amicizia, Laura Sticchi, Andrea Orsi, Giancarlo Icardi, Maria Francesca Piazza, Salini Mohanty, Francesca Senese, Gian Marco Prandi, Filippo Ansaldi

**Affiliations:** 1Regional Health Agency of Liguria (ALiSa), Piazza della Vittoria 15, 16121 Genoa, Italy; 2Department of Health’s Science (DiSSal), University of Genoa, 16132 Genoa, Italy; 3Hygiene Unit, San Martino Policlinico Hospital-IRCCS for Oncology and Neurosciences, 16132 Genoa, Italy; 4Merck & Co., Inc., 126 East Lincoln Ave., P.O. Box 2000, Rahway, NJ 07065, USA; 5MSD Italy, Via Vitorchiano 151, 00189 Rome, Italy

**Keywords:** pneumococcal disease, pneumonia, IPD, children, adolescents, adults, Italy

## Abstract

*Streptococcus pneumoniae* infection is responsible for significant morbidity and mortality, particularly in young children and older adults. The aim of this study was to investigate the incidence of hospitalisation and emergency department (ED) visits in relation to episodes of pneumococcal disease (PD) following the introduction of pneumococcal conjugate vaccines (PCVs) into the Liguria region of Italy. Between 2012 and 2018, episodes of all-cause pneumonia (80,152), pneumococcal-specific pneumonia (1254), unspecified pneumonia (66,293), acute otitis media (AOM; 17,040), and invasive PD (IPD; 1788) were identified from in-patient claims, ED and hospital discharge records, and the Liguria Chronic Condition Data Warehouse. In children < 15 years of age, pneumococcal pneumonia-related hospitalisations decreased from 35 to 13 per 100,000 person-years during the study period (*p* < 0.001); this decrease is potentially related to PCV use in children. All-cause pneumonia hospitalisations remained stable, whereas IPD hospitalisations increased and AOM hospitalisations decreased. In adults, hospitalisations for all-cause pneumonia increased from 5.00 to 7.50 per 1000 person-years (+50%; *p* < 0.001). Pneumococcal and unspecified pneumonia hospital admissions increased significantly during the study period, considerably affecting those ≥ 65 years of age. IPD hospitalisations varied across all age groups, but a significant change was not observed. Despite pneumococcal vaccination, substantial burden remains for PD in children and adults in Liguria, Italy.

## 1. Introduction

*S. pneumoniae* infections are a major cause of global morbidity and mortality, especially in vulnerable groups, such as older adults, children ≤ 5 years of age, and those at high risk of infection due to comorbidities [1,2]. *S. pneumoniae* infection can induce invasive and non-invasive pneumococcal disease and is the leading cause of community-acquired pneumonia worldwide [1,3,4].

The frequency of pneumococcal disease in infants has led to the introduction of pneumococcal conjugate vaccines (PCVs) across many regions, and a subsequent decline in vaccine-type paediatric invasive and non-invasive pneumococcal disease has been observed [5]. Young children who carry *S. pneumoniae* in the nasopharynx are also the main source of transmission to adults, in whom infection with *S. pneumoniae* can cause pneumococcal disease [6]. Reduction of vaccine serotype nasopharyngeal carriage through the use of PCVs in paediatric populations can indirectly provide protection to adults [6,7,8]. Alongside this, direct protection to older adults is available in the form of the 23-valent pneumococcal polysaccharide vaccine (PPSV23), which is available and recommended in many countries [9].

Pneumococcal disease is associated with substantial healthcare resource utilisation and direct medical costs [10,11,12]. As of 2018, the incidence of invasive pneumococcal disease (IPD) was on a rising trend in Europe, with the highest rates observed in those ≥ 65 years of age [13].

Liguria is an administrative region in the northwest of Italy and, as of 2020, has a population of approximately 1.52 million inhabitants [14]. In May 2003, a large-scale vaccination programme against pneumococcal disease was launched in this region, with all new-born children invited to receive 7-valent PCV (PCV7) according to a three-dose schedule (at 3, 5, and 11 months of age). This programme resulted in a rapid increase in PCV coverage to >80% in 2004 and >90% in 2007 across Liguria [15,16]. During the summer of 2010, PCV7 was replaced by 13-valent PCV (PCV13) according to a four-dose schedule (at 3, 5, 11, and 13 months of age, with a minimum gap of 2 months between doses), and one dose for children 2–5 years of age [16,17]. As of 2013, PCV vaccination coverage in children 24 months of age progressively increased for the 2005 to 2009 birth cohort, from 44.7% to 98.5% in 2011, respectively [18]. Owing to the reduction in vaccine-serotype circulation, the incidence of IPD in children 0–4 years of age decreased from 7.1 per 100,000 in 2008 to 3.8 per 100,000 in 2012 [19]. In 2020, national coverage was reported to be more than 90% among children 2 years of age belonging to the 2018 birth cohort [20].

In 2000, a similar large-scale pneumococcal vaccination programme was launched in adults in Liguria. This programme involved the recommendation of PPSV23 for older adults (≥64 years of age) or for adults of any age who have high-risk conditions or comorbidities. No vaccine coverage data are available regarding PPSV23 uptake in adults in Liguria; however, cumulative coverage rates calculated in a bordering region ranged from 26–31% in older adults and were approximately 23% in adults with high-risk conditions. In 2013, health authorities in Liguria recommended a pneumococcal vaccination series in adults, offering PCV13 followed by PPSV23 in adults 70–75 years of age [9].

This study investigated the incidence of hospitalisations due to IPD (pneumococcal and unspecified), unspecified pneumonia (conditions without pathogen attribution but in which *S. pneumoniae* is known to have a causative role), pneumococcal pneumonia, all-cause pneumonia (bacterial and viral causes), and acute otitis media (AOM) in children < 15 years of age. In addition, hospitalisation and emergency department (ED) visits associated with IPD, all-cause pneumonia, pneumococcal pneumonia, or unspecified pneumonia in adolescents ≥ 15 years of age and adults were studied.

## 2. Materials and Methods

### 2.1. Study Design and Population

This was a retrospective observational analysis that assessed the clinical burden of pneumococcal disease, including IPD (pneumococcal and unspecified), pneumococcal pneumonia, unspecified pneumonia, and all-cause pneumonia, in children, adolescents, and adults. AOM hospitalisations and ED visits were also measured in children from October 2012 to December 2018 in the Liguria region of Italy.

The Ligurian Regional Administrative Database covered the entire population residing in Liguria and included approximately 1.5 million inhabitants residing under five local health authorities. Patients < 15 years of age with one or more inpatient or ED claims for IPD, pneumonia (all-cause, pneumococcal, and unspecified), or AOM, and patients ≥ 15 years of age with one or more inpatient claims for pneumonia (all-cause, pneumococcal, and unspecified) or IPD (pneumococcal or unspecified), were identified.

Details included in the database were date of visit, information on diagnosis and procedures performed (according to International Classification of Diseases, Ninth Revision, Clinical Modification [ICD-9-CM] diagnosis codes), length of stay, date of discharge, and discharge status (i.e., died or discharged alive).

### 2.2. Episode Identification and Definition

For all patients included in the study, pneumonia episodes were identified using pneumococcal-specific ICD-9-CM codes, which indicated that the condition was caused by *S. pneumoniae*. In addition, codes were included for conditions without pathogen attribution but in which *S. pneumoniae* is known to have a causative role (unspecified). Furthermore, the definition of all-cause pneumonia comprised all other types of pneumonia, such as bacterial or viral pneumonia (Appendix A). IPD episodes were identified with pneumococcal-specific ICD-9-CM codes for pneumococcal bacteraemia, meningitis, bacteraemic pneumonia and other IPD, and unspecified invasive disease (including meningitis, bacteraemia, bacteraemic pneumonia, and other diseases) potentially caused by *S. pneumoniae*. For patients < 15 years of age, episodes of AOM were also identified through ICD-9-CM codes (Appendix A).

### 2.3. Study Endpoints

Episodes of all-cause pneumonia, pneumococcal-specific and unspecified pneumonia, AOM, and IPD (pneumococcal and unspecified) were identified from in-patient claims during the study period, plus ED and hospital discharge records, as well as the Liguria Chronic Condition Data Warehouse, as described above and in Appendix A.

The annual incidence of hospital admissions related to IPD and pneumonia was defined as the annual number of inpatient admissions per 1000 or 100,000 person-years.

### 2.4. Statistical Analysis

The clinical burden of pneumonia was stratified according to age, time period, and comorbidity risk factors, reporting the standard deviation for the mean age and frequency distributions with 95% confidence intervals (95% CIs) for the other categoric variables. Trends in incidence were assessed using the chi-squared test; a *p*-value of <0.05 was considered statistically significant.

Univariate analysis was used for relative risks of comorbidities for pneumococcal pneumonia and IPD. Fisher’s exact test was applied for those cases with less than five events in the population with or without a risk factor. Data were analysed using the JMP version 13.0.0 software (SAS Institute, Cary, NC, USA).

## 3. Results

### 3.1. Patient Demographics

In children < 15 years of age, a total of 9750 episodes of all-cause pneumonia were identified, including 287 episodes of pneumococcal pneumonia and 5964 episodes of unspecified pneumonia that were potentially caused by *S. pneumoniae*. A larger proportion of these episodes was observed in children 0–4 years of age compared with children 5–14 years of age (Table 1).

In addition, 17,040 episodes of AOM and 878 episodes of IPD and unspecified invasive disease potentially caused by *S. pneumoniae* were identified in children < 15 years of age, including 695 episodes of bacteraemia, 172 cases of other invasive pneumococcal diseases (e.g., pneumococcal septicaemia), and 11 episodes of meningitis. In children 0–4 years of age, 10,562 episodes of AOM, 578 episodes of bacteraemia, and nine episodes of meningitis were identified (Table 1).

In patients ≥ 15 years of age, a total of 70,402 episodes of all-cause pneumonia were identified; of these, 60,529 episodes were unspecified pneumonia and 967 were pneumococcal pneumonia. In addition, 1082 episodes of IPD were identified, including 129 episodes of meningitis and 953 episodes of bacteraemic pneumonia (pneumococcal-specific), pneumococcal septicaemia, pneumococcal infection, and empyema or pleural effusion (Table 2). Larger proportions of all-cause pneumonia (52,325 [74.3%] vs. 18,077 [25.7%]), pneumococcal pneumonia (695 [71.9%] vs. 272 [28.1%]), and unspecified pneumonia (47,308 [78.1%] vs. 13,284 [21.9%]) episodes were observed in older adults (>64 years of age) compared with individuals 15–64 years of age, respectively (Table 2).

### 3.2. Incidence of Hospitalisations and ED Visits

#### 3.2.1. Children < 15 Years of Age

The incidence of hospitalisation for all-cause pneumonia in children 0–14 years of age remained stable throughout the study period (4.67 per 100,000 person-years [95% CI: 4.25–5.00]). Pneumococcal pneumonia-related hospital admissions significantly decreased from 35 to 13 per 100,000 person-years (*p* < 0.001) between 2012 and 2018 (Figure 1A). In children < 2 years of age, the incidence fluctuated, with a range of 36–146 cases per 100,000 person-years (*p* = 0.53, R^2^ = 0.03). In children 2–4 and 5–14 years of age, the incidence significantly decreased from 2012 to 2018 (from 69 to 22 cases per 100,000 person-years [*p* < 0.01, R^2^ = 0.7] and from 21 to 8 cases per 100,000 person-years [*p* < 0.001, R^2^ = 0.7], respectively) (Table 3). Hospitalisations for unspecified pneumonia also decreased, although not significantly, in children < 15 years of age (Figure 1B).

Annual incidence of IPD hospitalisation in children < 15 years of age showed an increase over the study period, with values ranging from 1.70 cases per 100,000 person-years in 2012 to 4.85 cases per 100,000 person-years in 2018 (*p* < 0.001) (Table 3).

In addition, the annual incidence of AOM hospitalisation showed a decreasing trend, from 0.447 to 0.327 cases per 1000 person-years (*p* = 0.94) from 2012 to 2018, peaking in 2017 at 0.460 cases per 1000 person-years (Table 4). An increasing trend in the annual incidence of AOM ED visits, not followed by hospitalisation, was observed, from 6.506 to 7.208 cases per 1000 person-years (*p* = 0.34) from 2012 to 2018.

#### 3.2.2. Adolescents and Adults ≥ 15 Years of Age

The incidence of all-cause pneumonia, pneumococcal pneumonia, and unspecified pneumonia hospital admissions increased between 2012 and 2018 in all age groups but was more apparent in those 65–84 and ≥85 years of age. In adults diagnosed with all-cause pneumonia, an increase in the annual incidence of hospitalisation was observed, increasing from 5.00 to 7.50 per 100,000 person-years (+50%, *p* < 0.001; Figure 2A). The annual incidence rate of hospitalisation for all-cause pneumonia increased significantly in the 65–84 years age group (from 8.95 to 12.70 per 1000 person-years; +42%, *p* < 0.001) and in adults ≥ 85 years of age (from 26.63 to 38.58 per 1000 person-years; +45%, *p <* 0.001; Figure 3).

In those diagnosed with pneumococcal pneumonia, hospitalisation rates also increased significantly during the study period, from 0.5 to 1.2 per 10,000 person-years (+140%, *p* < 0.001; Figure 2B). Most notably, significant increases were observed in adults 65–84 years of age (from 0.10 to 0.22 per 1000 person-years; +120%, *p* < 0.001), whereas non-significant increases were seen in adults ≥ 85 years of age (from 0.21 to 0.45 per 1000 person-years; +114%, *p* = 0.25; Appendix A).

Similarly, the annual incidence rate of hospitalisation for unspecified pneumonia also increased significantly, from 4.0 to 6.0 per 1000 person-years (+49%, *p* < 0.001; Figure 2C). In particular, significant increases were seen in adults 65–84 years of age (from 7.35 to 10.24 per 1000 person-years; +39%, *p* < 0.001) and adults ≥ 85 years of age (from 25.25 to 36.38 per 1000 person-years; +44%, *p* < 0.001; Appendix A).

The overall incidence of IPD hospitalisation did not increase significantly over time, showing high variability in all age groups analysed. However, IPD incidence was significantly higher in the older (65–84 and ≥85 years) age groups (Appendix A).

Throughout the study period, the incidence of meningitis and other disease manifestations (pneumococcal-specific bacteraemic pneumonia, pneumococcal septicaemia, pneumococcal infection, and empyema or pleural effusion) remained relatively stable. Meningitis comprised 11.0% (range 7.2–13.0) of total IPDs, and other disease manifestations comprised 89.0% (range 87.0–92.8).

### 3.3. Analysis of Comorbidities

#### Adolescents and Adults ≥ 15 Years of Age

The prevalence of comorbid conditions in the Liguria population demonstrated an increasing trend with age. Generally, the incidence of hospitalization for all-cause pneumonia was highest in patients with chronic respiratory illness and chronic cardiovascular disease (Figure 3).

Across most age groups in the adult population, the relative risk of contracting pneumococcal-related pneumonia in patients with comorbidities was statistically significant in comparison with patients without a comorbidity, with the highest relative risk observed in patients 15–19 years of age with diabetes (115.82, 95% CI: 12.05–1112.71; *p* = 0.011) (Table 5).

The hospitalisation rates for IPD among adults with comorbidities 65–84 and ≥85 years of age were also numerically higher than in adolescents and adults < 65 years of age during the study period (2012–2018).

## 4. Discussion

The aim of this study was to investigate the incidence of hospitalisation and ED visits in relation to episodes of pneumococcal disease following the introduction of PCVs into the Liguria region of Italy. In this retrospective analysis, there was an observed decrease in annual pneumonia-related hospital admissions for children < 15 years of age between 2012 and 2018. Overall, a greater decrease in pneumococcal pneumonia hospital admissions was observed compared with unspecified pneumonia admissions, with similar rates observed for children < 5 years of age and those 5–14 years of age. The decrease in the incidence of pneumococcal-related hospitalisation observed in this study corroborates findings from other trials and observational studies that report a decrease in a similar trend in IPD and pneumonia episodes in young children, following the successful implementation of PCV13 vaccination in infants [21,22]. However, within the decreasing trend observed in our study, there was an anomaly in 2016, when the incidence of hospitalisation was significantly lower in comparison with other years assessed. There are two potential factors that could contribute to this outlier. First could be the update to the indications contained in the Ligurian Vaccine Prevention Plan, implemented at the start of 2016, which offered the pneumococcal vaccine free of charge to high-risk subjects of all ages [23]. This could have led to a decrease in the incidence of severe disease and explain the low values observed in 2016. A second variable that could have led to the decrease in incidence could be related to the low circulation of influenza-like illness and symptoms reported in the 2015–2016 season, which was the lowest peak recorded between 2012 and 2018 [24].

In addition, there was a decreasing trend observed in AOM hospitalisation and ED visits for children < 15 years of age, which supports the findings of other studies in Italian regions. During 2002 to 2014 in Tuscany, Boccalini et al. observed a decrease in hospitalisations due to AOM in children and an increase in hospitalisations for potentially associated pneumococcal diseases in older subjects > 64 years of age [25].

In contrast to the decreasing trend in AOM hospitalisation, a generally stable trend in IPD hospitalisations was observed in the analysis, with a slight overall increase in cases in children < 5 years of age, peaking in 2017.

In adults and adolescents, the incidence of all-cause pneumonia-related hospitalisation increased across all age groups throughout the study period but was more apparent in those ≥ 65 years of age. Unspecified pneumonia comprised the majority (~80%) of pneumonia cases.

In adults and adolescents, the incidence of IPD hospitalisation showed an increasing trend across the study period but was not significant, apart from in the group ≥ 65 years of age. This increase in cases is mirrored in a national Italian study that analysed general IPD incidence from 2007–2017, highlighting the largest increase in IPD observed in individuals >64 years of age [26]. Despite this general increasing trend in IPD observed in our study, high variability across all the adult age groups was observed; this could be due to two potential factors. Age groups with a low incidence of IPD hospitalisation may receive indirect protection against pneumococcal disease following the introduction of PCVs into paediatric immunisation programmes [27]. In age groups in which a high incidence in IPD hospitalisation was observed, the phenomenon of immunosenescence, associated with older age, could increase the disease burden linked with IPD, leading to a higher incidence of hospitalisation [28]. Comorbidities of a respiratory, renal, or cardiovascular nature were associated with an increase in the burden of pneumonia and IPD in adults ≥ 65 years of age, supporting the rationale for vaccine recommendation in older adults.

The findings of this study are likely to be a direct result of the introduction and uptake of PCV13 vaccination in infants in the Liguria region [21]. However, despite the decrease in incidence of pneumococcal pneumonia-related hospital admissions reported here, pneumonia is still a major burden in children globally [29]. Similar to this, the clinical burden of ED access and hospitalisations due to pneumonia has increased in adolescents and adults since the introduction of PCV13 in 2010; yet, of these episodes, pneumococcal pneumonia only accounted for approximately 1% of all pneumonia cases. However, inaccurate diagnosis could have contributed to the small number of cases, which may not accurately reflect the true incidence of pneumococcal pneumonia. This highlights the current unmet need to distinguish between *S. pneumoniae* pathogens colonising the upper respiratory tract and *S. pneumoniae* causative of pneumonia [30].

The results of our study confirm that the burden of pneumococcal disease remains substantial, particularly in older age groups, even with under-reporting and under-coding by hospital discharge records. In order to attain broader protection against pneumococcal disease and reduce the burden in the adult population, it would be necessary to strike a balance between maintaining disease reduction for current vaccine serotypes, while expanding the additional serotype coverage through research and development.

### 4.1. Strengths of the Study

One strength of this study is that it considered multiple clinical manifestations of pneumonia, including both IPD and pneumococcal-specific pneumonia episodes in addition to unspecified pneumonia. This study also provides two estimates of the clinical burden of pneumonia: one that is highly specific, with a degree of certainty that episodes included in the analysis are caused by *S. pneumoniae*, and one that would capture additional undetected episodes of pneumococcal pneumonia associated with other pathogens, describing instances of co-infection. Furthermore, our analysis is based on real-world data from the Liguria Regional Administrative Database, including and assessing numerous records on hospitalisation and ED visits.

### 4.2. Limitations of the Study

Within the study design, there is a limitation regarding the assignment of diagnosis codes, such as ICD-9-CM. The assignment of these codes does not necessarily indicate disease presence, due to the possibility of incorrect coding. A further limitation is the uncertainty of the aetiology of the diseases that required hospitalisation, making it difficult to fully attribute the increasing trends in the hospitalisation to pneumococcal disease. However, hospital discharge records that describe the final diagnosis of a patient could be a reliable proxy for the circulation of *S. pneumoniae* in the Ligurian population.

## 5. Conclusions

Following the introduction of PCV13 into the infant immunisation programme throughout Italy since 2010, the annual incidence of pneumococcal pneumonia-related hospitalisation decreased significantly in children < 15 years of age. However, the general incidence of all-cause and pneumococcal pneumonia significantly increased in patients 65–84 and >85 years of age in the Liguria region of Italy. This reinforces the belief that, despite protection in children through the use of PCVs, and subsequent indirect protection in adults, there is an unmet need for increased vaccination uptake in older adults. The impact of new and future PCVs on the incidence of pneumococcal disease will depend on the proportion of disease caused by *S. pneumoniae* and the prevalence of vaccine-type serotypes in children and adults, substantiating the view that further PCV development with a wider range of serotypes is also warranted.

## Figures and Tables

**Figure 1 vaccines-10-01375-f001:**
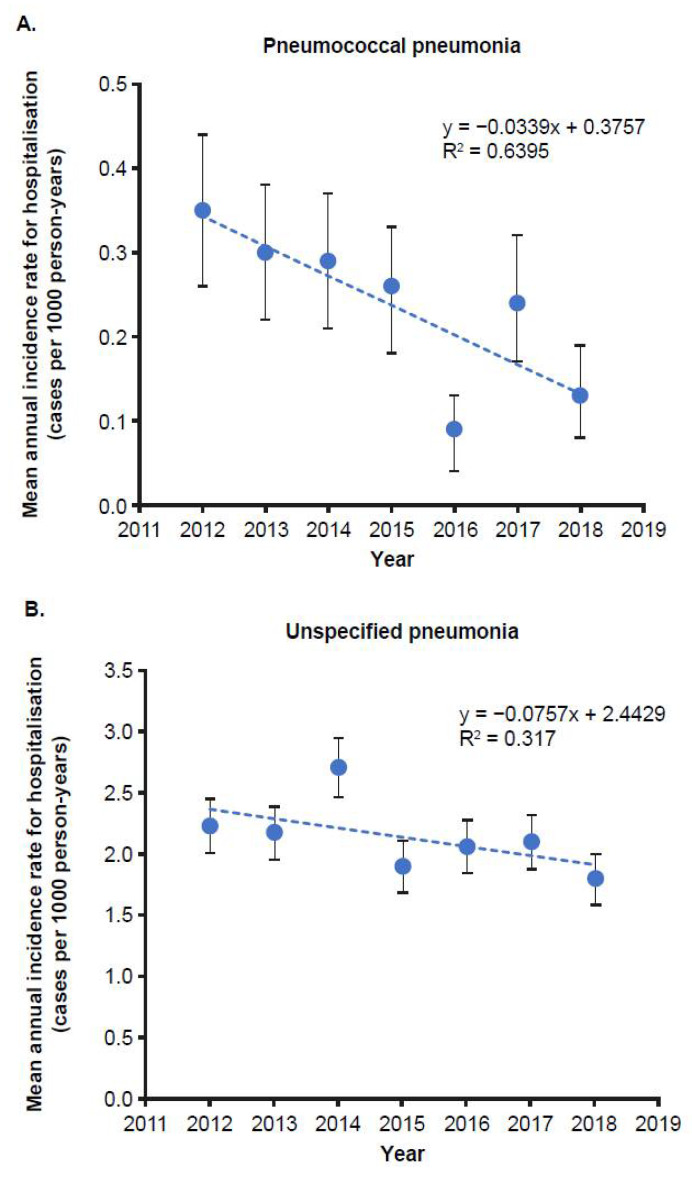
Annual incidence of (**A**) pneumococcal pneumonia and (**B**) unspecified pneumonia hospitalisation in Liguria, Italy, in children < 15 years of age.

**Figure 2 vaccines-10-01375-f002:**
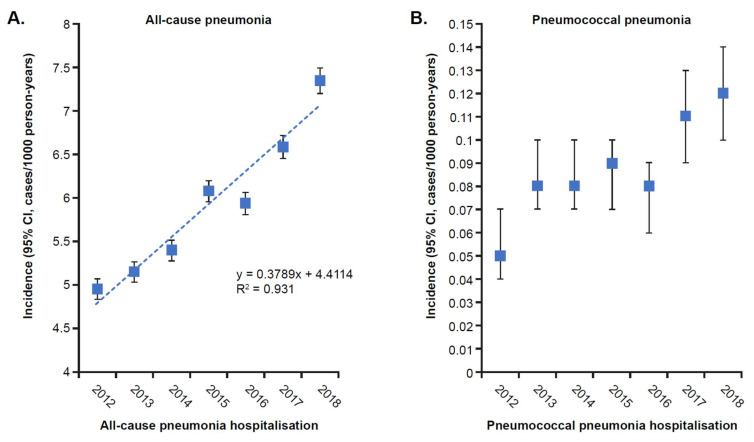
Annual incidence of (**A**) all-cause, (**B**) pneumococcal, and (**C**) unspecified pneumonia hospitalisation in adolescents and adults ≥ 15 years of age in Liguria, Italy (2012–2018). CI: confidence interval.

**Figure 3 vaccines-10-01375-f003:**
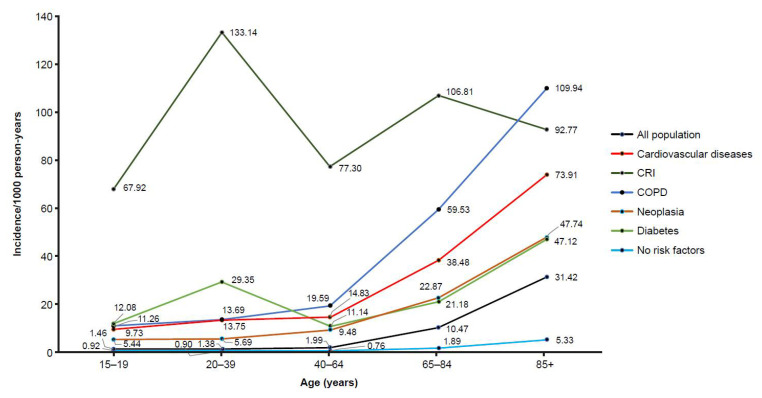
Mean incidence of hospitalisation for all-cause pneumonia in adolescents and adults ≥ 15 years of age in Liguria, Italy (2012–2018), stratified by comorbidities. COPD: chronic obstructive pulmonary disease; CRI: chronic respiratory illness.

**Table 1 vaccines-10-01375-t001:** Demographic characteristics and risk factors for invasive and non-invasive pneumonia episodes in children < 15 years of age (2012–2018).

	All-Cause Pneumonia (*n* = 9750)	Pneumonia		IPD				Otitis Media (*n* = 17,040)	Other ^†^ (*n* = 172)	Total (*n* = 27,668)
				Meningitis		Bacteraemia				
		Pneumococcal (*n* = 287)	Unspecified (*n* = 5964)	Pneumococcal (*n* = 5)	Unspecified (*n* = 6)	Pneumococcal (*n* = 629)	Unspecified (*n* = 66)			
Male, *n* (%)	5130 (52.6)	150 (52.4)	3081 (51.7)	3 (60.0)	5 (83.3)	368 (58.5)	35 (52.6)	9385 (55.1)	117 (67.8)	15,030 (54.3)
Age, mean (SD)	5.25 (3.6)	4.26 (2.9)	5.02 (3.3)	4.05 (1.4)	1.74 (2.7)	2.70 (2.9)	1.70 (3.4)	4.76 (3.4)	8.72 (4.4)	5.01 (3.6)
Age group, *n* (%)										
0–4 years	5555 (57.0)	194 (67.7)	3507 (58.1)	4 (80.0)	5 (83.3)	518 (82.4)	60 (91.2)	10,562 (62.0)	38 (22.4)	16,485 (59.6)
5–14 years	4195 (43.0)	93 (32.3)	2457 (41.2)	1 (20.0)	1 (16.7)	111 (17.7)	6 (8.8)	6478 (38.0)	134 (77.6)	11,118 (40.4)
Previous PCV, *n* (%)	>90%	>90%	>90%	>90%	>90%	>90%	>90%	>90%	>90%	>90%
Comorbidities, *n* (%)	2205 (22.6)	64 (22.3)	1255 (21.1)	2 (33.3)	1 (8.3)	110 (17.5)	7 (10.5)	1528 (9.0)	25 (14.5)	3184 (11.5)
Renal	10 (0.1)	1 (0.3)	7 (0.1)	0	0	1 (0.2)	0	12 (0.07)	0	20 (0.1)
Cardiovascular	329 (3.4)	23 (8.2)	214 (3.6)	2 (33.3)	1 (8.3)	53 (8.5)	5 (7.0)	361 (2.1)	12 (7.2)	659 (2.4)
Respiratory	1633 (16.8)	52 (18.2)	1003 (16.8)	0	0	72 (11.5)	1 (1.8)	762 (4.5)	11 (6.6)	20,807 (7.5)
Diabetes	21 (0.2)	0	3 (1.0)	0	0	1 (0.1)	0	12 (0.1)	0	28 (0.1)
Neoplasia	93 (1.0)	8 (2.8)	49 (0.8)	0	0	23 (3.6)	0	70 (0.4)	0	155 (0.6)
Other	296 (3.0)	9 (3.1)	175 (2.9)	0	0	17 (2.7)	1 (1.8)	398 (2.3)	1 (0.7)	624 (2.3)

**^†^** The “Other” category comprised pneumococcal-specific conditions including pneumococcal arthritis, pneumococcal peritonitis, and the following disease manifestations with and without the presence of pneumococcal infection: infective pericarditis, acute and subacute bacterial endocarditis, acute and subacute endocarditis, spontaneous bacterial peritonitis, acute or unspecified osteomyelitis, and pyogenic/unspecific arthritis. IPD: invasive pneumococcal disease; PCV: pneumococcal conjugate vaccine; SD: standard deviation.

**Table 2 vaccines-10-01375-t002:** Demographic characteristics and risk factors for invasive and non-invasive pneumonia episodes in adolescents and adults ≥ 15 years of age (2012–2018).

	Pneumonia			IPD	
	Pneumococcal (*n* = 967)	Unspecified (*n* = 60,529)	All-Cause (*n* = 70,402)	Meningitis (*n* = 129)	Others ^†^ (*n* = 953)
Male, *n* (%) (95% CI)	513; (53.05) [49.91–56.2]	32,380; (53.44) [53.04–53.84]	37,830; (53.73) [53.37–54.1]	63; (48.84) [40.21–57.46]	562; (58.97) [55.85–62.09]
Age, mean (SD)	73.20 (16.63)	75.23 (16.83)	72.80 (18.76)	62.55 (14.89)	67.85 (20.30)
Age group, *n* (%) (95% CI)					
15–64 years	272; (28.13) [25.29–30.96]	13,284; (21.92) [21.59–22.25]	18,077; (25.68) [25.35–26.00]	63; (48.84) [40.21–57.46]	282; (29.59) [26.69–32.49]
>64 years	695; (71.87) [69.04–74.71]	47,308; (78.08) [77.75–78.41]	52,325; (74.32) [74.00–74.65]	66; (51.16) [42.54–59.79]	671; (70.41) [67.51–73.31]
Comorbidities, *n* (%) (95% CI)	786; (81.28) [78.82–83.74]	49,872; (82.31) [82.00–82.61]	56,705; (80.54) [80.25–80.84]	87; (67.44) [59.36–75.53]	832; (87.30) [85.19–89.42]
Renal failure	130; (13.44) [11.29–15.59]	10,508; (17.34) [17.04–17.64]	11,386; (16.17) [15.90–16.44]	6; (4.65) [1.02–8.29]	181; (18.99) [16.50–21.48]
Cardiovascular	469; (48.50) [45.35–51.65]	34,603; (57.11) [56.71–57.5]	38,109; (54.13) [53.76–54.50]	36; (27.91) [20.17–35.65]	529; (55.51) [52.35–58.66]
Chronic respiratory disorders	255; (26.37) [23.59–29.15]	16,599; (27.39) [27.04–27.75]	19,734; (28.03) [27.70–28.36]	10; (7.75) [3.14–12.37]	149; (15.63) [13.33–17.94]
Diabetes	184; (19.03) [16.55–21.5]	11,686; (19.29) [18.97–19.60]	13,408; (19.04) [18.75–19.33]	16; (12.40) [6.71–18.09]	190; (19.94) [17.40–22.47]
Neoplasia	204; (21.10) [18.52–23.67]	12,096; (19.96) [19.64–20.28]	13,759; (19.54) [19.25–19.84]	20; (15.50) [9.26–21.75]	304; (31.90) [28.94–34.86]
Other	370; (38.26) [35.20–41.33]	22,611; (37.32) [36.93–37.70]	25,517; (36.24) [35.89–36.60]	31; (24.03) [16.66–31.40]	375; (39.35) [36.25–42.45]

^†^ Bacteraemic pneumonia (pneumococcal-specific), pneumococcal septicaemia, pneumococcal infection, and empyema or pleural effusion. CI: confidence interval; IPD: invasive pneumococcal disease; SD: standard deviation.

**Table 3 vaccines-10-01375-t003:** Annual hospitalisation for pneumococcal-specific pneumonia and IPD between 2012–2018, by age cohort.

Age, Years	Year	Cases Per 100,000 Person-Years(95% CI)	
		Pneumococcal-Specific Pneumonia	IPD
0–14	2012	35 (26–44)	1.70 (−0.22–3.62)
	2013	30 (22–38)	2.27 (0.05–4.50)
	2014	29 (21–37)	4.01 (1.04–6.98)
	2015	26 (18–33)	2.32 (0.05–4.60)
	2016	9 (4–13)	3.53 (0.71–6.36)
	2017	24 (17–32)	7.17 (3.11–11.22)
	2018	13 (8–19)	4.85 (1.49–8.20)
0–1	2012	79 (30–128)	0.00 (0.00–0.00)
	2013	39 (5–74)	0.00 (0.00–0.00)
	2014	74 (26–123)	0.00 (0.00–0.00)
	2015	94 (39–150)	8.55 (−8.21–25.31)
	2016	36 (1–71)	8.97 (−8.61–26.55)
	2017	146 (75–218)	18.26 (−7.05–43.57)
	2018	47 (6–88)	9.41 (−9.03–27.86)
2–4	2012	69 (43–96)	5.33 (−2.06–12.71)
	2013	60 (35–86)	8.24 (−1.08–17.56)
	2014	74 (45–102)	0.00 (0.00–0.00)
	2015	52 (28–77)	5.83 (−2.25–13.90)
	2016	15 (2–28)	5.96 (−2.30–14.23)
	2017	34 (14–54)	18.61 (3.72–33.51)
	2018	22 (6–39)	9.57 (−1.26–20.41)
5–14	2012	21 (13–28)	0.79 (−0.76–2.34)
	2013	20 (13–28)	0.79 (−0.76–2.33)
	2014	12 (6–18)	5.51 (1.43–9.59)
	2015	12 (6–18)	0.79 (−0.76–2.35)
	2016	5 (1–9)	2.40 (−0.32–5.11)
	2017	11 (5–17)	3.22 (0.06–6.38)
	2018	8 (3–13)	3.25 (0.07–6.43)

CI: confidence interval; IPD: invasive pneumococcal disease.

**Table 4 vaccines-10-01375-t004:** Annual incidence of AOM hospitalisation and ED visits between 2012 and 2018, for children < 15 years of age.

Year	Cases Per 1000 Person-Years (95% CI)	
	AOM (Hospitalisation)	AOM (ED Visits)
2012	0.447 (0.348–0.545)	6.506 (6.131–6.880)
2013	0.403 (0.309–0.497)	7.184 (6.790–7.579)
2014	0.418 (0.322–0.514)	6.671 (6.289–7.053)
2015	0.337 (0.250–0.424)	7.457 (7.050–7.863)
2016	0.418 (0.321–0.516)	6.822 (6.431–7.214)
2017	0.460 (0.357–0.563)	6.642 (6.253–7.031)
2018	0.327 (0.240–0.440)	7.208 (6.800–7.616)

AOM: acute otitis media; CI: confidence interval; ED: emergency department.

**Table 5 vaccines-10-01375-t005:** Relative risk of comorbidities of pneumococcal pneumonia and IPD in adolescents and adults ≥ 15 years of age in Liguria, Italy (2012–2018).

Age (Years)	Pneumococcal Pneumonia		Invasive Pneumococcal Disease	
	Relative Risk (95% CI)	*p*-Value	Relative Risk (95% CI)	*p*-Value
Chronic obstructive pulmonary disease				
15–19	7.39 (0.77–71.00)	0.16 ^†^	0.00 (–)	–
20–39	6.39 (2.70–15.09)	<0.01	2.97 (0.71–12.45)	0.15 ^†^
40–64	6.91 (4.56–10.49)	<0.01	4.22 (2.26–7.85)	<0.01
65–84	7.40 (6.04–9.05)	<0.01	2.72 (1.83–4.04)	<0.01
≥85	5.41 (4.10–7.15)	<0.01	4.28 (2.13–8.60)	<0.01
Cardiovascular diseases				
15–19	0.00 (–)	–	0.00 (–)	–
20–39	7.78 (1.89–32.13)	<0.01	5.77 (0.79–42.36)	0.16 ^†^
40–64	6.22 (4.26–9.09)	<0.01	11.84 (7.95–17.64)	<0.01
65–84	5.90 (4.88–7.15)	<0.01	3.22 (2.41–4.30)	<0.01
≥85	10.11 (7.10–14.38)	<0.01	7.79 (3.58–16.95)	<0.01
Diabetes				
15–19	115.82 (12.05–1112.71)	0.011 ^†^	0.00 (–)	–
20–39	4.50 (0.62–32.68)	0.20 ^†^	22.01 (6.68–72.55)	<0.01 ^†^
40–64	8.13 (5.59–11.82)	<0.01	3.58 (1.92–6.67)	<0.01
65–84	1.89 (1.50–2.38)	<0.01	1.72 (1.20–2.45)	<0.01
≥85	2.04 (1.49–2.80)	<0.01	1.83 (0.84–3.97)	0.12
Chronic renal injuries				
20–39	20.33 (2.80–147.51)	0.05 ^†^	0.00 (–)	–
40–64	8.52 (3.77–19.23)	<0.01	15.66 (7.29–33.64)	<0.01
65–84	5.40 (4.04–7.21)	<0.01	5.88 (3.86–8.95)	<0.01
≥85	6.24 (4.68–8.32)	<0.01	5.12 (2.49–10.5)	<0.01
Neoplasia				
15–19	0.00 (–)	–	0.00 (–)	–
20–39	2.04 (0.28–14.83)	0.39 ^†^	0.00 (–)	–
40–64	2.96 (1.84–4.77)	<0.01	4.55 (2.75–7.52)	<0.01
65–84	3.35 (2.74–4.11)	<0.01	3.63 (2.69–4.91)	<0.01
≥85	1.39 (0.97–1.99)	0.08	1.09 (0.43–2.80)	0.80 ^†^

All data were calculated by univariate analysis. ^†^ Application of Fisher’s exact test for <5 cases reported. CI: confidence interval; IPD: invasive pneumococcal disease.

## Data Availability

See http://www.icmje.org/recommendations/browse/publishing-and-editorial-issues/clinical-trial-registration.html (accessed on 18 June 2022). Merck Sharp & Dohme LLC, a subsidiary of Merck & Co., Inc., Rahway, NJ, USA’s data sharing policy, including restrictions, is available at http://engagezone.msd.com/ds_documentation.php (accessed on 9 November 2021) through the EngageZone site or via email to dataaccess@merck.com.

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
