# Peer review of "Incidence of Hospitalisation and Emergency Department Visits for Pneumococcal Disease in Children, Adolescents, and Adults in Liguria, Italy: A Retrospective Analysis from 2012–2018"

_vaccines, 2022, doi:10.3390/vaccines10091375_

Round 1

Reviewer 1 Report

In this manuscript, Astengo et al., describe the incidences of pneumococcal disease and hospitalisation in Liguria, Italy. This retrospective analysis should be of significant interest to epidemiologists, clinicians, and vaccine research. I have few comments that needs to be addressed.

1) In Fig. 1A, what are the reasons for the incidence rates to be lower in the year 2016? This needs to be discussed. A age stratified patient demographics and no. of samples would add to the scope of this manuscript

2) All statistical significances are by univariate analysis. Did the authors perform multivariate analysis adjusted for comorbidities variables such as in Table 5? any significance or lack significance obtained from this multivariate analysis should be discussed

3) The authors could add how their incidence rates and clinical data overlap or vary with the other published literatures.

Reviewer 2 Report

Dear Authors.

I consider the manuscript submitted by you to be of very high quality clinical, scientific, and community interest. For which I believe it is necessary to make improvements in the tables and figures. However, the text of the document explains and resolves the questions raised at the beginning of the manuscript.

My comments regarding the abstract are that it describes the situation in a very good way, however, it does not make explicit in a good way the question that the work you present pursues, which is explicitly stated in the discussion section.

Regarding the methodologies and results, the descriptions described are very good and go to the point of interest. However, the tables and figures that are relevant to the development of the idea expressed in the document, do not contribute anything and are not discussed in a good way. For example, table N°1 has the separation between 0-4 years of age, and from 5-14 years, however, the document only mentioned the data or figures for children under 15 years, and so on with the remaining tables and figures. If the authors showed these data in the tables, it is so that the reader can see the use that the authors make of this data, and be able to better understand the context that the data have in the main question to be answered by the work. 

The discussion section is very agile and concise in its development, leaving the reader with the feeling that this is a study that is useful for specialists and health professionals who work with these data.

Reviewer 3 Report

good work deserves publication

Author Response

Thank you for your response. 

Round 2

Reviewer 2 Report

Dear authors.

Thank you for add my comments to manuscript. I agree with changes.

Congratulations.